Can endocranial volume be estimated accurately from external skull measurements in great-tailed grackles (Quiscalus mexicanus)?

Logan Corina J. 1 * cl417@cam.ac.uk
Palmstrom Christin R. 2
1 SAGE Center for the Study of the Mind, University of California Santa Barbara , Santa Barbara, CA , USA
2 College of Creative Studies, University of California Santa Barbara , Santa Barbara, CA , USA
Druzinsky Robert
* Current affiliation: Department of Zoology, University of Cambridge, Cambridge, UK

Electronic publication date: 2015 Jun 11
Publication date: 2015
Volume: 3
Electronic Location ID: e1000
Received 2015 Jan 13; Accepted 2015 May 14
Copyright: © 2015 Logan et al.
Copyright year: 2015
Copyright holder: Logan et al.
License: This is an open access article distributed under the terms of the Creative Commons Attribution License, which permits unrestricted use, distribution, reproduction and adaptation in any medium and for any purpose provided that it is properly attributed. For attribution, the original author(s), title, publication source (PeerJ) and either DOI or URL of the article must be cited.
License URL: https://creativecommons.org/licenses/by/4.0/

Keywords: Endocranial volume, Great-tailed grackle, Quiscalus mexicanus, Bird, Skull measurements, Bead method

Funding: National Geographic Society/Waitt Grants Program W252-12 Undergraduate Research and Creative Activities at UCSB This research was funded by the National Geographic Society/Waitt Grants Program, the SAGE Center for the Study of the Mind (CJL) and the Undergraduate Research and Creative Activities grant (CP) at the University of California Santa Barbara. The funders had no role in the study design, data collection and analysis, decision to publish, or preparation of the manuscript.

==============================
There is an increasing need to validate and collect data approximating brain size on individuals in the field to understand what evolutionary factors drive brain size variation within and across species. We investigated whether we could accurately estimate endocranial volume (a proxy for brain size), as measured by computerized tomography (CT) scans, using external skull measurements and/or by filling skulls with beads and pouring them out into a graduated cylinder for male and female great-tailed grackles. We found that while females had higher correlations than males, estimations of endocranial volume from external skull measurements or beads did not tightly correlate with CT volumes. We found no accuracy in the ability of external skull measures to predict CT volumes because the prediction intervals for most data points overlapped extensively. We conclude that we are unable to detect individual differences in endocranial volume using external skull measurements. These results emphasize the importance of validating and explicitly quantifying the predictive accuracy of brain size proxies for each species and each sex.

Introduction

While comparing relative brain sizes (corrected for body size) across species has led to a greater understanding of the evolutionary factors correlated with brain size at a broad taxonomic scale (e.g., Iwaniuk & Nelson, 2003; Sakai et al., 2011; Sol et al., 2005), little is known about the within species causes and consequences of variation in brain sizes (see Gonda, Herczeg & Merilä, 2013; Thornton & Lukas, 2012). Additionally, the accuracy of brain size proxies, which are frequently used in such comparisons, is not often validated (Healy & Rowe, 2007). Therefore, the accuracy of brain size estimations and how they compare to estimations in other species is questionable (Healy & Rowe, 2007). Intraspecific brain size comparisons are rare perhaps due to the difficulty of obtaining data on a number of factors for the same individuals (e.g., biometric measurements, reproductive success, dominance rank, position in the social network, and cognitive abilities). Acquiring such data is key for understanding what contributes to the evolution of brain size among individuals, as well as across species (Gonda, Herczeg & Merilä, 2013; Logan & Clutton-Brock, 2013; Thornton & Lukas, 2012).

We investigated whether endocranial volume, a proxy for brain size (Iwaniuk & Nelson, 2002), can be approximated using measurements of the external skull in great-tailed grackles (Quiscalus mexicanus, JF Gmelin, 1788). Grackles are invasive, having successfully expanded their geographical range by exploiting new environments (Peer, 2011). Invasion success is considered a measure of behavioral flexibility and positively correlates with relative brain size across bird species (Sol et al., 2002). One of us (Logan) is conducting in depth investigations of great-tailed grackle cognition and behavioral flexibility in the lab and field to understand whether invasiveness directly indicates behavioral flexibility and what kinds of cognition grackles use to solve novel problems. This study system provides a rare opportunity to examine intraspecific differences in brain size, which could have implications for understanding range expansions in the context of behavioral flexibility. Finding an accurate proxy for endocranial volume would greatly ease the collection of data on brain sizes since external skull measurements could be further validated to account for head measurements that can be taken on live birds, thus allowing for correlations with a number of other factors on which data are gathered on this species.

This study is intended as a first step in validating the accuracy of using head measurements taken from live birds to predict their actual brain size, which would require two additional steps: (1) validating the link between external skull measurements and external head measurements, and (2) validating the link between endocranial volume and actual brain size. Regarding the latter validation, there is reason to believe that endocranial volume accurately approximates actual brain volume in birds because the meninges, the matter between the skull and the brain, are thinner than in mammals and the shape of the braincase in birds tightly tracks the shape of the brain (Iwaniuk & Nelson, 2002). Additionally, it was previously observed in the genus Quiscalus that individual variation in endocranial volume is particularly large when compared with variation in other bird species (Quiscalus quiscala; Iwaniuk & Nelson, 2002). Assuming great-tailed grackles have similar amounts of variation, our study provides an opportunity to understand how much of this variation in endocranial volume is due to variation in skull morphology (i.e., changes in skull length, width, and/or height specifically).

Great-tailed grackle body sizes are sexually dimorphic (Johnson et al., 2000), therefore we expected sex differences in endocranial volumes and we investigated proxies for each sex independently. We used endocranial volumes calculated from computerized tomography (CT) scans to represent actual endocranial volumes since this measure is the most precise. The complete area of the inside of the skull is accounted for in CT scans, while other methods do not cover the whole endocranial space (Witmer et al., 2008; Knoll et al., 2012). We compared CT volumes to skull length, width, and height measurements to determine whether the correlation between these two methods and the accuracy of external skull measures in predicting CT volumes warrants their use as a proxy for endocranial volume. We also evaluated the bead method of measuring endocranial volume, where glass beads are poured into the skull and then out into a graduated cylinder, to increase the value of our research by determining whether this widely used method (e.g., Isler et al., 2008; Iwaniuk & Nelson, 2002) also accurately predicts actual endocranial volume as estimated by CT scans in this species.

Methods

Specimens

We collected data from February through September 2014, and in March 2015, on 40 great-tailed grackle skulls (Table S1), 20 female and 20 male (some analyses have 19 males because on one of their skulls the bill was broken off, thus we could not acquire its skull length measurement), obtained from the Museum of Southwestern Biology (n = 24, Albuquerque, NM), the Ornithology Division of the University of Kansas (KU) Biodiversity Institute (n = 15, Lawrence, KS), and the Santa Barbara Museum of Natural History (n = 1, Santa Barbara, CA). Skulls of unknown age were aged by Andy Johnson if they were from the Museum of Southwestern Biology or by us if they were from KU. Skulls were aged using the percentage of ossification to classify each as adult (>7 months old; 100% ossified unless it was collected in February–May because this would mark the start of that individual’s first breeding season after having hatched June-August in the previous year) or immature (<7 months old; <100% ossified when collected September–December indicating it had hatched that year; del Hoyo, Elliot & Christie, 1992; Winker, 2000; Pyle, Howell & DeSante, 1997).

Collecting endocranial volume measurements

Linear measurements

Linear measurements of skulls were collected by placing calipers in locations on the skull that would also be accessible and measurable on a live bird in the field. We recorded skull length from the base of the bill to the back of the skull along the occipital crest (Fig. 1A), height from the posterior edge of the foramen magnum (the posterior edge of the neck at the base of the skull on a live bird) to the top of the skull along the frontal region (Fig. 1B), and width at the widest part of the braincase along the squamosal bones (Fig. 1C). All measurements were taken to the nearest 0.1 mm. Research on other species has found positive correlations between actual brain mass and brain volume estimated from linear skull measurements by calculating the volume of an ellipsoid (barn swallows: Møller, 2010), and endocranial volume and the volume of a cube as estimated using head width (zebra finches: Bonaparte, Riffle-Yokoi & Burley, 2011). Therefore, we estimated endocranial volume using a number of volumetric shapes and data transformations to determine which best correlated with endocranial volumes from CT scans. The volumetric shapes included were: cube (Length × Width × Height), sphere (43πr3, where r=12L or 12W or 12H), ellipsoid (43πabc, where a=12L, b=12W, c=12H), and cone/pyramid (13bh, where b = W, h = H). We included log, natural log, and exponential transformations of the data, and also allowed polynomial terms.

Figure 1 Skull length, height, and width.

Measuring skull length (A), skull height (B), replicating the height that can be measured on live birds, and skull width (C) at the widest part of the braincase.

CT scans

Skulls were CT scanned at the Pueblo Radiology Medical Group in Santa Barbara, California using a Siemans 16-slice Somatom Sensation 16 (1 mm slices, 100 Kv, 150 MAs, 380 mm FOV, soft tissue window, analyzed with bone algorithm on). Endocranial volume (cm3) was calculated using the DICOM viewer OsiriX v5.8.5 (32-bit, Pixmeo SARL, Switzerland; Figs. S1 and S2) for 1 mm cross-sectional slices (regular) and for 1 mm cross-sectional slices that were taken with the CT scanner bed moved 0.5 mm forward (offset), using the average endocranial volume regular+offset2 in analyses. The offset was added to increase the precision of the endocranial volume measurements since grackle craniums are small (approximately 20 mm in length), resulting in about 20 slices per scan (one slice every 1 mm). The offset allowed us to measure more area (one slice every 0.5 mm) by increasing the number of slices to approximately 40 per skull.

Beads

Endocranial volume was measured by pouring 1 mm diameter glass beads (BioSpec Products, catalog number 11079110) into the cranium through the foramen magnum until full. The skull was repeatedly shaken to settle the beads and then filled again until the beads reached the posterior foramen magnum without falling out (Fig. 2). The volume was calculated by pouring the beads out of the skull and into a graduated cylinder (5 ml in 0.1 ml graduations, World Precision Instruments, Inc., Sarasota, Florida, USA, catalog number CG-0160; note that 1 ml = 1 cm3). In cross-species comparisons, there is mixed evidence about whether pouring the beads into a graduated cylinder introduces error when compared with pouring the beads onto a scale and converting their mass into volume (4% difference: Miller, 1997, 0% difference: Isler et al., 2008). The skulls in this study were measured with an average error of 2% (i.e., there was a 2% difference in volume between two sets of volume measurements carried out on a subset of 12 skulls), which is small in comparison to the variance between skulls (intra-class correlation coefficient = 0.94; Hutcheon, Chiolero & Hanley, 2010). Therefore, the error should not affect the power to detect a correlation with the more precise CT method (intra-class correlation coefficient = 0.98) of measuring skulls.

Figure 2 Bead method.

Skull holes are plugged with cotton and then the cranium is filled with glass beads.

Statistical analyses

The female and male data (analyzed separately) were normally distributed (Anderson Darling normality test: females: skull height p = 0.27, length p = 0.30, width p = 0.86; males: skull height p = 0.35, length p = 0.63, width p = 0.38). We defined statistical significance as p < 0.05 throughout the paper. Two sets of linear, bead, and CT scan measurements were taken on a subset of skulls on different days by Palmstom to quantify the random measurement error (intraobserver reliability). We used intra-class correlation coefficients (ICC) to determine the precision of our estimates using the equation in Fig. 2 in Hutcheon and colleagues (2010): true slope×variance true X valuesvariance true X values + variance random error (we assumed that variance in the observed values was equal to the variance in the true values). This ICC is a measure of consistency, not agreement since it does not include rater effects (Auerbach, La Porte & Caputo, 2004).

We used generalized linear models (GLMs) to determine how well linear and bead measurements correlated with volumes from CT scans, while examining whether the age of the individual at death and the year the skull was collected improved the model fit. GLMs were carried out in R v3.1.2 (R Core Team, 2014) using the MCMCglmm function (MCMCglmm package; Hadfield, 2010), while applying the dredge function (MuMIn package; Barton, 2012) to select the best-fitting model using the Akaike weight (Akaike, 1981; Burnham & Anderson, 2002). We considered the best-fitting model to be strongly supported and reliable if its Akaike weight was ≥0.9 as suggested by Burnham & Anderson (2002) since this would indicate that the likelihood of the model given the data is very high. Female and male data were analyzed in separate models. Full models included (1) endocranial volumes from CT scans as the response variable with the following explanatory variables: volume of a cube or sphere or ellipsoid or cone + age and the interaction with year collected; or (2) endocranial volumes from CT scans as the response variable with the following explanatory variables: skull length + skull width + skull height + age and the interaction with year collected. GLMs were conducted on the best-fitting model for each sex to explore whether the adjusted coefficient of determination (adjusted r2) improved by transforming the endocranial volume proxy (explanatory variable) in the following ways: squared, cubed, quadratic, exponential, square root, log, log base 10, and a polynomial with a degree of two or three. Of these, the model with the highest adjusted r2 was chosen as the final best-fitting model for that sex and included in the results below. Interpretations of correlation strengths were taken from Taylor (1990): ≤0.35 = weak correlation, 0.36–0.67 = moderate, 0.68–0.89 = high, 0.90–1.00 = very high. We set the minimum criteria for a correlation of sufficient strength such that it might be predictive at r2 ≥ 0.88, which is equivalent to Pearson’s r set to alpha = 0.05 or r ≥ 0.95, adjusted for the random measurement error in the response variable (CT measurements), which has an intra-class correlation coefficient of 0.99 (0.95∗0.99 = 0.94, 0.942 = 0.88; Hutcheon, Chiolero & Hanley, 2010).

Since we want to predict CT volumes from linear skull measures, we validated whether this was possible by generating prediction intervals with 95% confidence levels. We applied the predict function in the MCMCglmm package to the best-fitting model for each sex and evaluated whether fitted values (predicted CT volumes) had credible intervals small enough such that there was little to no overlap with other fitted values, thus allowing the discrimination of individual differences.

Results

Intraobserver reliability

There was very high within-method consistency (precision) between the two sets of CT and bead volume measurements, but no consistency for linear volume (LWH) measurements when sexes were analyzed together and separately (Table 1). There was high (sexes combined) to very high (sexes analyzed separately) consistency when comparing the two sets of skull width measurements, high consistency for skull length (when sexes were separate and combined), and moderate (males and sexes combined) to very high (females) consistency for skull height (Table 1).

Table 1 Intraobserver reliability.

Summary data and results.

Variable	Measurement 1	Measurement 2	ICC	
	Female	Male	Both	Female	n	Male	n	Both	Female	Male	Both	
Volume CT	2.30 ± 0.15	2.56 ± 0.06	2.43±0.17	2.38 ± 0.17	4	2.48 ± 0.08	4	2.38 ± 0.16	0.99	0.99	0.95	
Volume LWH	13,814 ± 1,415	15,605 ± 1,276	14,650 ± 1,417	13,779 ± 1,233	5	15,320 ± 785	7	14,672 ± 1,218	0.009	0.003	0.003	
Volume bead	2.72 ± 0.15	2.93 ± 0.15	2.88 ± 0.21	2.68 ± 0.19	5	2.91 ± 0.13	7	2.84 ± 0.21	0.92	0.94	0.94	
Skull Length	29.04 ± 0.98	30.06 ± 0.81	29.72 ± 0.99	29.40 ± 1.27	5	30.24 ± 0.78	7	29.94 ± 1.00	0.83	0.76	0.82	
Skull Width	23.83 ± 0.86	23.99 ± 0.36	23.83 ± 0.67	23.39 ± 0.90	5	24.01 ± 0.39	7	23.81 ± 0.69	0.92	0.92	0.88	
Skull Height	20.24 ± 0.87	20.89 ± 1.44	20.65 ± 1.18	19.96 ± 0.48	5	21.09 ± 0.54	7	20.68 ± 0.76	0.91	0.51	0.65	
Notes.

Measurements mean ± standard deviation

Both data from both sexes combined

ICC intra-class correlation

Units of measurementCT cm3

bead ml

LWH mm3

length/width/height mm

Correlations between methods

None of the models using linear measurements to explain variation in CT volumes were likely given the data, as indicated by the low Akaike weights of the best-fitting models (<0.90; Table 2; Burnham & Anderson, 2002). Regardless, we used the best-fitting models to examine these relationships further. The volume of a sphere was the best-fitting shape for both sexes (the radius was based on skull width for males and skull height for females). The best-fitting female model showed a positive relationship between CT volumes and volumes from using the skull height as the radius for a sphere, volumes were larger for immatures than for adults, and volumes slightly decreased over the years collected (Table 2, model 1; Fig. 3A). The best-fitting male model showed a positive correlation between CT volumes and volumes using a quadratic polynomial of the skull width as the radius for a sphere, volumes were slightly larger for immatures than for adults, and volumes decreased slightly over the years collected (Table 2, model 2; Fig. 3B). Transformations of the explanatory volume variables or substituting volume for individual linear measurements (length, width, height, or some combination of these) did not improve the adjusted r2 for females. Volumes from CT scans were moderately (males) to highly (females) correlated with spherical volumes from linear measurements (Table 2).

Figure 3 Plots of the volumes of spheres and volumes calculated from CT scans for females and males.

Correlations between CT volumes and the volume of a sphere as calculated from linear measurements for female (A) and male (B) adults (small circles) and immatures (large circles), with the year the skull was collected represented by a red-blue spectrum (earlier years are redder and recent years are bluer). Note that regression lines only reflect the relationship between spherical volume and CT volume and do not correct for age or year (factors in the best-fitting model for both sexes) as in the GLMs. Skulls were aged as described in the methods.

Table 2 Model results.

Outputs for the best-fitting female and male models from dredge in R. Note that these models were the best-fitting relative to other models not shown here and the models here cannot be compared with each other.

Method	Sex	Model	Akaike weight	Adjusted r2	p	y =	
CT-sphere	Female	1	0.60	0.80	<0.0001	0.00002 × VolumeSphere + 1.10 × Age − 0.007 × Year + 1.44	
Male	2	0.26	0.39	0.02	0.47 × VolumeSphere + 0.19 × VolumeSphere2 + 0.12 × Age − 0.003 × Year + 5.25	
CT-bead	Female	3	0.74	0.77	<0.0001	0.37 × VolumeBead − 0.0005 × Year + 11.24	
Male	4	0.45	0.68	<0.0001	0.66 × VolumeBead − 0.09 × Age + 0.66	

None of the models using bead volumes to explain variation in CT volumes were strongly supported given the data as indicated by their low Akaike weights for the best-fitting models (<0.90; Table 2; Burnham & Anderson, 2002). Nonetheless, the best-fitting female model showed that endocranial volumes decreased slightly over time (Table 2, model 3; Fig. 4A), while the best-fitting male model included age, with immatures having smaller volumes than adults (Table 2, model 4; Fig. 4B). Volumes from CT scans were highly positively correlated with volumes from the bead method for both sexes (Table 2).

Figure 4 Plots of the bead volumes and CT volumes for females and males.

Correlations between CT volumes and bead volumes for female (A) and male (B) adults (small circles) and immatures (large circles), with the year the skull was collected represented by a red-blue spectrum (earlier years are redder and recent years are bluer). Note that regression lines only reflect the relationship between bead volume and CT volume and do not correct for age (in the best-fitting male model) or year (in the best-fitting female model) as in the GLMs. Skulls were aged as described in the methods.

None of the correlations between CT volumes and linear measures met our minimum criteria (r2 ≥ 0.88) for a strong enough relationship such that they might predict endocranial volumes from the linear measurements of skulls. Since we want to predict CT volumes from linear measures, we determined whether this was possible by generating prediction intervals for the best-fitting female and male models for the linear measurements (models 1 and 2) and bead method (models 3 and 4; Table 2). We found that the lower and upper limits of the 95% credible intervals of the predicted values for both sexes show extensive overlap such that individual differences would not be able to be resolved if a new, unvalidated data point was obtained (Table 3).

Table 3 Prediction analysis results.

Predicted CT volume (fitted value) and the predicted intervals (with lower and upper bounds) in which these new data points would occur with 95% credible intervals based on inputs from linear measures or the bead method in the best-fitting female and male models for each method.

Linear measurements	Bead method	
Males	Females	Males	Females	
Fitted value	Lower	Upper	Fitted value	Lower	Upper	Fitted value	Lower	Upper	Fitted value	Lower	Upper	
2.72	2.37	3.07	2.23	2.03	2.42	2.61	2.42	2.80	2.30	2.07	2.48	
2.41	2.17	2.73	2.08	1.86	2.29	2.43	2.23	2.61	2.21	1.99	2.42	
2.40	2.12	2.69	1.87	1.61	2.07	2.37	2.18	2.55	1.90	1.63	2.11	
2.60	2.33	2.92	1.99	1.78	2.22	2.70	2.52	2.90	2.02	1.78	2.26	
2.42	2.10	2.70	2.20	2.02	2.39	2.50	2.34	2.68	2.05	1.81	2.30	
2.52	2.22	2.83	2.30	2.10	2.51	2.55	2.35	2.75	2.42	2.22	2.62	
2.77	2.45	3.07	2.43	2.23	2.60	2.76	2.54	2.94	2.39	2.16	2.59	
2.67	2.41	2.94	2.39	2.20	2.58	2.70	2.50	2.89	2.28	2.07	2.52	
2.35	2.06	2.64	2.32	2.14	2.53	2.37	2.17	2.56	2.31	2.11	2.54	
2.55	2.30	2.84	2.51	2.30	2.71	2.50	2.31	2.67	2.57	2.31	2.81	
2.46	2.21	2.72	2.51	2.30	2.71	2.43	2.25	2.63	2.43	2.20	2.64	
2.64	2.40	2.92	2.39	2.19	2.59	2.63	2.45	2.82	2.32	2.10	2.53	
2.53	2.24	2.77	2.48	2.27	2.70	2.44	2.26	2.63	2.46	2.20	2.70	
2.58	2.31	2.85	2.40	2.21	2.60	2.37	2.18	2.57	2.32	2.11	2.52	
2.55	2.31	2.85	2.32	2.13	2.52	2.47	2.28	2.67	2.31	2.11	2.56	
2.64	2.39	2.94	2.41	2.23	2.60	2.76	2.57	2.97	2.43	2.20	2.64	
2.54	2.29	2.82	2.36	2.16	2.55	2.61	2.40	2.80	2.43	2.22	2.66	
2.55	2.24	2.80	2.42	2.21	2.60	2.67	2.49	2.87	2.43	2.20	2.65	
2.52	2.26	2.80	2.32	2.13	2.52	2.48	2.27	2.68	2.36	2.15	2.57	
2.51	2.24	2.80	2.02	1.82	2.13	2.57	2.38	2.74	1.98	1.74	2.18	

Comparing method means

Endocranial volume means were significantly different from each other when comparing across methods (mean ± standard deviation: females: CT 2.29 cm3 ± 0.20, sphere 32,459.1 mm3 ± 4,344.7, bead 2.60 ml ± 0.28; males: CT 2.54 cm3 ± 0.15, sphere 59,292.1 mm3 ± 2,360.4, bead 2.91 ml ± 0.21; Welch two sample t-test: females: CT x Sphere t = 31, p < 0.0001, df = 19; sphere x bead t = − 31, p < 0.0001, df = 19; bead x CT t = 4, p = 0.0003, df = 34; males: CT x sphere t = 96, p < 0.0001, df = 19; sphere x bead t = − 96, p < 0.0001, df = 19; bead x CT t = 6, p < 0.0001, df = 35).

Discussion

While female great-tailed grackle endocranial volumes from linear measurements were highly correlated with volumes from CT scans, which we consider a more accurate proxy for brain size than bead volume (Witmer et al., 2008; Knoll et al., 2012), the correlation did not meet our criteria of having a coefficient of determination (r2) equal to or greater than 0.88—a level of correlation that might be strong enough to allow for the resolution of individual differences in endocranial volumes. This correlation was only moderate in males, which is likely due to the sexual dimorphism in this species. Our sample includes individuals from a range of populations in which the extent of sexual dimorphism might vary. If, in some populations, there has been strong selection for males to increase in body size, then the skeletal measures will not reflect brain size accurately because, in many instances, skeletal size changes faster than brain size as has been found in primates (Montgomery, 2011). Perhaps additional biometric measurements would explain more of the variation in their endocranial volumes from CT scans; however, we only had access to skulls for most of the specimens and therefore could not test this hypothesis.

We were more interested in whether a given value of an external skull measurement could accurately predict actual endocranial volume from CT scans, rather than setting a subjective criterion about how high r2 should be, especially given the extensive debate around the latter approach (e.g., Legates & McCabe Jr, 1999; Müller & Büttner, 1994). In particular, r2 “[…] describes the proportion of the total variance in the observed data that can be explained by the model” (Legates & McCabe Jr, 1999, p. 233, emphasis added) and thus does not allow one to investigate differences in the variance of individual data points. Our predictive analyses showed that prediction intervals for new data points overlapped to such a degree (within 95% credible intervals) that it was not possible to distinguish among individuals, as we would need to when collecting linear measurements on new individuals in the field. We must conclude that external skull measurements are not accurate enough to estimate endocranial volume in great-tailed grackles.

Predictive analyses are crucial for determining the accuracy of predicting individual data points by a particular method and should be applied extensively in future research, rather than relying solely on correlation coefficients (r) or coefficients of determination (r2). The omission of such an analysis leaves data uninterpretable for its purported use of discerning intraspecific differences in a morphological feature. Additionally, we caution against using a proxy validated in one species as evidence that the same proxy will apply to other species (e.g., great tits: Dreyer, 2012). Until intraspecific validations of brain size proxies using skull or head measurements have been validated across species, we cannot assume that what works (or not) for one species will work (or not) for another.

The bead method was highly correlated with CT volumes in both sexes, however, it also did not meet our minimum criteria (r2 ≥ 0.88), and prediction intervals extensively overlapped for individual data points. Great-tailed grackles and common grackles are among the species with the largest ranges in endocranial volumes (as measured using the bead method) when compared with most other species in Iwaniuk & Nelson’s (2002) study on 81 bird species. Both grackle species had large standard deviations when compared with mean volumes (common grackles: mean ± SD = 2.59 ml ± 0.37, n = 10, Iwaniuk & Nelson, 2002; great-tailed grackles: female 2.60 ml ± 0.28, n = 20, male 2.91 ml ± 0.21, n = 20, this study). It appears that grackle endocranial volumes are more variable than those of many other species. This is not likely due to variation in skull morphology since we did not find a perfect correlation between endocranial volume and external skull measurements. Therefore, we caution against using external skull measurements to estimate endocranial volume without proper validation.

To infer differences in brain size among individuals of the same species, and of the same sex, there must be a high degree of accuracy to have the ability to detect actual individual differences (Legates & McCabe Jr, 1999; Logan & Clutton-Brock, 2013). Our results highlight the need to validate brain size and/or endocranial volume proxies and their predictive power for each species under investigation, and for each sex if they are sexually dimorphic. It is unfortunate that there is not an easier, more accurate way to approximate brain size in the field where we have the potential to understand how evolutionary factors drive brain size variation within species. However, this study accentuates the importance of knowing how accurate brain size measures and proxies are when including such data in analyses.

Supplemental Information

Figure S1 Skulls in the CT scanner

Grackle skulls on the CT scanner bed about to be scanned.

Click here for additional data file.

Figure S2 CT scan of grackle skulls

Screenshot of a CT scan showing five grackle skulls using the software OsiriX.

Click here for additional data file.

Table S1 Skull data

Archive data for each Quiscalus mexicanus skull measured (SBMNH, Santa Barbara Museum of Natural History; MSB, Museum of Southwestern Biology, KU, University of Kansas Biodiversity Institute and Natural History Museum). ∗, Quiscalus mexicanus.

Click here for additional data file.

We thank Scott Grafton at the University of California Santa Barbara (UCSB) and Lawrence Harter and Chuck Scudelari at the Pueblo Radiology Medical Group for providing access to the CT scanner; Mario Mendoza at UCSB for scan planning; Sabrina Cruz and Kerri Moore at the Pueblo Radiology Medical Group for scanning; Steve Rothstein for advice and sponsoring CP’s grant; and Mireia Beas-Moix and the Cheadle Center for Biodiversity and Ecological Restoration at UCSB for hosting the skull loans; Dieter Lukas for analysis assistance; and Jay Taylor and an anonymous reviewer for feedback on a previous version of this manuscript. We are grateful for skull loans from the Museum of Southwestern Biology (and to Andy Johnson for aging their skulls), Krista Fahy and the Santa Barbara Museum of Natural History, and the Ornithology Division of the KU Biodiversity Institute.

Additional Information and Declarations

Competing Interests

Author Contributions

Animal Ethics

Data Deposition

The authors declare there are no competing interests.

Corina J. Logan conceived and designed the experiments, analyzed the data, contributed reagents/materials/analysis tools, wrote the paper, prepared figures and/or tables, reviewed drafts of the paper.

Christin R. Palmstrom performed the experiments, contributed reagents/materials/analysis tools, reviewed drafts of the paper.

The following information was supplied relating to ethical approvals (i.e., approving body and any reference numbers):

No approval was necessary since we used skulls from museum collections.

The following information was supplied regarding the deposition of related data:

The data is online at the knb data repository (https://knb.ecoinformatics.org/#view/doi:10.5063/F1668B3W; Logan & Palmstrom, 2015).

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
