# Peer review of "Can endocranial volume be estimated accurately from external skull measurements in great-tailed grackles (Quiscalus mexicanus)?"

_PeerJ, doi:10.7717/peerj.1000_

## Round 0.1 · original submission · Major Revisions

Dear Dr. Logan,
Since both reviewers had very serious concerns, please be sure that you address each issue directly in your resubmission.
Sincerely,
Robert

·

Basic reporting

It is regretted, however, that its structure and data of contents are poorly performed in order to be publishable (The details are as follows).

Experimental design

I think the authors adopted cumbersome methodology for the purpose of “find the proxy for brain size” and it can be more simplified. I could not understand why the authors transform the linear measurements to some volumetric shapes: cube, sphere, ellipsoid, con/pyramid. What did they base that on? Using the multiple regression analysis would be a good way to detect or assess the appropriate proxy for endocranial volume. The formula for the brain volume could be obtained by the multiple regression analysis (Kawabe et al.:2009 Brain Behav Evol 74, 295–301).

Some previous study mentioned the relation between the CT and bead method, and the authors should cite those studies (e.g. Ashwell & Scofield, 2008: Brain Behav Evol 71, 151–166). The authors acquired CT images from medical machine. However, obtaining the images of avian skulls by using the medical CT is inappropriate. The resolution is too coarse for passerines. I do not recommend the measurement of this precision. I think the bead method is enough for this study, rather than using medical CT.

Validity of the findings

Although the data seems to be properly obtained, no raw data was shown in the text. It makes difficult for readers to validation of this study. Additionally, the results were improperly-shown and it is very hard to understand the foundation for the authors’ conclusions. In addition, a large part of the results section is redundant. If the data and results are appropriately arranged in tables and figures, the problem can be helped a lot. I would encourage the authors to rewrite the results section with a colleague familiar with morphometrics.

As mentioned above, it is necessary for the authors to show raw data of measurements. We cannot validate the reproducibility and/or findings without raw data.

L. 150–189: The authors might arrange the statistics in tables and figures.

Additional comments

The authors investigated the relationship between endocranial volume and skull measurements in great-tailed grackles. Although they found moderately significant relationships between them, failed to significant correlations between endocranial volumes and linear measurements. They concluded that the brain volume cannot be estimated from external skull measurements. However, as the authors also stated in the text, acquiring intraspecific data of brain volume is so important, and the intention of authors is highly valued.

Reviewer 2 ·

Basic reporting

General format conforms to the standards with the following critical exceptions:
--The title needs to be modified to accurately reflect the scope of the study. The study did not investigate "brain" size, but endocranial size. In addition, it did not examine field proxies (which would be measurements of live or intact specimens), but instead, skeletal specimens.
--The Introduction should state why this particular species was used to investigate the broader question being addressed in the study. What makes the species ideal for the study?
--Lack of citation or evidence to support a number of claims made by the authors (see specific comments in attached document).
--Figures are in serious need of consolidation, editing, and formatting (i.e., Figs. 1–3 should be cropped and combined as parts of single figure; Fig. 4 should be cropped; Figs. 4, 8 are not necessary for the narrative of the study (can be included as supp. info.); Figs. 6, 7 should be combined; I see no reason for the vertical axes in Figs. 6, 7 to be as long as they are in their current state).

Experimental design

Although the objective of the study (i.e., to identify potential proxies for endocranial/"brain" size) is of interest to many biologists (including myself), I think that the experimental design is fundamentally flawed for a number of reasons:
--The manuscript fails to provide justifications for why this particular bird species is a good subject for this study. Even if the conclusions presented here are true, their extensibility to other bird species or a broader group is lacking (as authors points out in the Discussion section). One is left wondering, "Why do this study at such limited scope?"
--Despite its objective, the study does not use brain size, but endocranial size as proxy for brain size without testing. Although previous studies have reported a high degree of congruence between endocranial and brain size in birds (e.g., Hopson 1979; Iwaniuk & Nelson 2002; Striedter 2005), it is difficult to justify equating endocranial and brain sizes for this study without any tests because the assumption directly underlies the principal objective of the study.
--Study claims testing for "field proxy" for "brain" size, but measurements on skull specimens defined by osteological markers (e.g., foramen magnum) do not reflect measurements that could be taken from live or intact specimens.
--Only three measurements were taken from the skull. Why these three? Why not additional ones?
--No justification given for estimating volumes of shapes (cube, sphere, ellipsoid, and cone/pyramid). Why not simply perform statistical analyses on raw morphometric measurements and endocranial volume?
--Correlation test used for assessing intraobserver error seems illogical. A correlation test assumes that there is no correlation between two variables (presumed to be between 1st and 2nd measurements), but we expect there to be a very high congruence between the two sets of measurements taken from the same specimens. Thus, a correlation test does not have an appropriate null and the low p-value is not surprising, but expected even with a fairly large amount of intra-observer error. One usually tests for the differences between two corresponding measurements to assess measurement error.
--The manuscript does not define what is considered a "significant" or "reliable" result in many cases.
--The "subjective minimum criteria" (Ln 174) of R^2 > 0.88 for a "strong relationship" is not justified on scientific or statistical grounds. This is poor scientific practice.
--The study fails to examine relative brain size (normalized for body size/mass), which is the standard metric used in comparative studies, not absolute brain size, which clearly correlates with body size of individuals.

References:
--Hopson, J. A. 1979. Paleoneurology; pp. 39–146 in C. Gans, R. G. Northcutt, and P. Ulinski (eds.), Biology of the Reptilia Vol. 9. Academic Press, New York, NY.
--Iwaniuk, A. N., and J. E. Nelson. 2002. Can endocranial volume be used as an estimate of brain size in birds. Canadian Journal of Zoology 81: 16–23.
--Striedter, G. F. 2005. Principles of Brain Evolution. Sinauer, Sunderland, MA.

Validity of the findings

Although the methods are fundamentally flawed, I find the result that simple skull measurements are poor proxies for endocranial volume convincing, at least for the grackle species examined here. However, I find it difficult to draw any broader implications for the study. More importantly, it fails to answer the principal aim of the study, which is to identify possible field proxies for brain size because (1) the "field proxies" are actually osteological proxies (i.e., not on live or feathered specimens); (2) the study does not exhaustively measure the skull (i.e., only three basic measurements); (3) endocranial size, instead of actual brain volume, is used, which is a close, but not a perfect proxy for brain size. The latter point is crucial for the primary aim of the study. In addition, the authors claim that the study supports the importance of examining intraspecific relationships between "brain" size and potential field proxies across multiple species, but the study does not support this statement unless the authors add a comparative aspect to the study showing differences in the pattern between multiple species (i.e., perform the same analyses on at least another bird species).

Additional comments

As I mentioned above, the question being investigated is of interest to many researchers. However, there are fundamental issues in the methods used that are logically flawed or are inconsistent with the principal objective of the study. Despite my decision to reject this manuscript in its current form, I also see great value in the study and would love to see it published. Therefore, I highly recommend that the authors resubmit a revised version that addresses most, or preferably all, of the critical issues raised above. Please also see the attached PDF for more specific and generally more minor comments.

Annotated reviews are not available for download in order to protect the identity of reviewers who chose to remain anonymous.

---

## Round 0.2 · Major Revisions

Thank you for your resubmission. The reviewer still has some major concerns. Please review the comments carefully and respond to each one. I would also like to add that you should be more explicit in with regard to relationship between endocranial volume and brain size and morphology, since your manuscript is based not only on finding useful proxies for endocranial volume, but also on endocranial volume as a proxy for brain size.

·

Basic reporting

The manuscript has been basically properly revised and became straightforward. It’s regretted, however, that it brings to light the potential problems of this study.

The authors rewrite the aim of this study, focusing on the estimation of the volume of endocranial cavity. However, that makes the potential value of the data obtained by the authors force down. Despite the existence of those valuable intraspecific data, they gave little discussion about the deviation of linear measurements and endocranial volume. I think it would be a waster and it is better to focusing on the individual differences than to the estimation of the endocranial volume. In addition, I don’t think that the estimation of the volume is reasonable motivation for this study.

Experimental design

No Comments

Validity of the findings

No Comments

Additional comments

L.62-64: I understand the authors attempt to explore the background this study, but I feel this sentence is over explanation and redundant.

L.67-74: As stated above, it’s a little bit of a stretch to set the aim to estimate the endocranial volume.

L.77-79: I think it is better to focus on this subject.

L. 96-100: Because they repeat that CT value is most the most precise throughout this manuscript, I don’t understand why the authors attempt to compare the value of CT volume and bead volume after all that. I feel the discussion about the comparison between two methods separated from other discussion in this study.

L.151: I read the response of the authors without believing that the resolution of CT data is well high. I think the scan subject (=skulls: approx. 20 mm) are too small for FOV 380 mm. Please show real CT images.

L.187-188 & L. 393-397: I suggest make a table which shows at a glance.

L. 407- 410 & Fig. 3: If the statement (i.e. skeletal size changes faster than brain size) is true, the plots of large circles (= immature) must be plotted relatively below in Figure 3.

---

## Round 0.3 · Minor Revisions

Dear Corina and Christin,

Thank you for the careful responses that you made in response to the review. Before your article is accepted, I ask that you add the image of the CT scan that was in the original manuscript and was removed in the revised submission. Please include it as a Supplementary Figure. Although the reviewer thought that it is not necessary, I am certain that many readers would like to have it available.

---

## Round 0.4 · accepted · Accept

Thank you for making all of the changes requested.